# Port Site Metastasis in Women with Low- or Intermediate-Risk Endometrial Carcinoma: A Systematic Review of Literature

**DOI:** 10.3390/cancers16152682

**Published:** 2024-07-27

**Authors:** Antonio Raffone, Diego Raimondo, Alessio Colalillo, Arianna Raspollini, Daniele Neola, Antonio Travaglino, Virginia Vargiu, Luigi Carlo Turco, Maria Giovanna Vastarella, Renato Seracchioli, Francesco Fanfani, Luigi Cobellis, Francesco Cosentino

**Affiliations:** 1Department of Woman, Child, and General and Specialized Surgery, University of Campania “Luigi Vanvitelli”, 80138 Naples, Italy; anton.raffone@gmail.com (A.R.); mariagiovanna.vastarella@studenti.unicampania.it (M.G.V.); luigi.cobellis@unicampania.it (L.C.); 2Department of Medical and Surgical Sciences (DIMEC), University of Bologna, 40138 Bologna, Italy; renato.seracchioli@unibo.it; 3Division of Gynecology and Human Reproduction Physiopathology, IRCCS Azienda Ospedaliero-Universitaria di Bologna, 40138 Bologna, Italy; 4Division of Gynecologic Oncology, Department of Woman and Child Health and Public Health, Fondazione Policlinico Universitario A. Gemelli IRCCS, Università Cattolica del Sacro Cuore, 00168 Rome, Italy; colalillo1987@gmail.com (A.C.); francesco.fanfani@policlinicogemelli.it (F.F.); 5Department of Neuroscience, Reproductive Sciences and Dentistry, School of Medicine, University of Naples Federico II, 80138 Naples, Italy; daniele.neola@unina.it; 6Unit of Pathology, Department of Medicine and Technological Innovation, University of Insubria, 21100 Varese, Italy; antonio.travaglino@uninsubria.it; 7Gynecologic Oncology and Surgery Unit, Responsible Research Hospital, 86100 Campobasso, Italy; virginia.vargiu@policlinicogemelli.it (V.V.); francesco.cosentino@unimol.it (F.C.); 8Department of Medicine and Health Science “V. Tiberio”, University of Molise, 86100 Campobasso, Italy; 9Ovarian Cancer Center, Candiolo Cancer Institute, FPO-IRCCS, 10060 Turin, Italy; luigicarlo.turco@ircc.it

**Keywords:** endometrial cancer, laparoscopy, robotics

## Abstract

**Simple Summary:**

We systematically reviewed the literature on port site metastases in low- and intermediate-risk endometrial carcinoma. We searched six electronic databases and compared data on PSM from the included studies, finding seven studies and 13 cases of PSM in patients with low- or intermediate-risk endometrial carcinoma. Local resection and radiotherapy appear to be the most appropriate treatment approaches. Unfortunately, no consensus has been reached about treatment, and the prognosis appears poor.

**Abstract:**

**Background:** Port site metastasis (PSM) has been reported as a rare metastasis in women with endometrial carcinoma (EC). However, even more rarely, it has also been described in patients with low- or intermediate-risk EC. Unfortunately, knowledge appears limited on the topic. **Objectives:** Our objective was to systematically review the literature on PSM in low- or intermediate-risk EC. **Search Strategy:** A systematic review of the literature was performed by searching six electronic databases from their inception to January 2023. **Selection Criteria:** We included in our research all peer-reviewed studies which reported PSM in low- or intermediate-risk EC women. **Data Collection and Analysis:** Data on PSM were collected from the included studies and compared. **Results:** Seven studies with 13 patients (including our case) were included in the systematic review. PSM was reported in patients with low- or intermediate-risk EC independently from tumor histologic characteristics, endoscopic approach, lymph node staging type, number and site of the port, route of specimen removal, prevention strategies for PSM, and concomitant metastases. Among several proposed treatments, local resection and radiotherapy with or without chemotherapy might be the most appropriate ones. Nevertheless, the prognosis appears poor. **Conclusions:** In patients with low- or intermediate-risk EC, PSM can occur as a rare metastasis, regardless of tumor characteristics or surgical strategy. Unfortunately, no consensus has been reached regarding treatment, and the prognosis appears poor. Additional cases are needed in order to confirm and further explore this rare EC metastasis.

## 1. Introduction

The role of minimally invasive surgery for gynecological cancer treatment has increased over the last decades. Although these new techniques have revolutionized the gynecological oncological field, allowing for shortened hospital stays, lower postoperative complications, and a higher quality of life compared to laparotomy, focus must be placed on the possibility of metastases at the laparoscopic port site [1].

Port site metastases (PSM) are defined as “early tumor recurrences that develop locally in the abdominal wall, within the scar tissue of one or more trocar sites or an incision wound after laparoscopy, and are not associated with peritoneal carcinomatosis” [2].

In 1978, Dobronte et al. first described a local metastasis where a pneu-needle was inserted for a diagnostic laparoscopy for ovarian cancer; this metastasis was explained by the authors by the presence of ascites containing tumor cells [3]. For gynecological tumors, the mean incidence of PSM is reported to be between 1% and 2%, with the highest incidence among ovarian cancer patients (19.6%) [4,5].

Although PSM’s etiology is still unclear, it is considered to be multifactorial: some of the proposed PSM mechanisms are port site contamination secondary to a sub-optimal surgical technique or leakage of aerosolized tumor cells through the port site (“chimney effect”), and local immune reactions occur due to gas exposition used for laparoscopy [6].

Endometrial carcinoma (EC) is the most prevalent gynecologic tumor in the Western world, and the fourth-most frequent one in women worldwide, with increasing incidence and mortality in the last few decades [7,8,9,10]. In EC women, PSM’s incidence is reported to be around 0.18–0.33%, mainly concerning high-risk cases [4,11,12]. However, PSM can rarely occur in low- or intermediate-risk EC patients. Unfortunately, to the best of our knowledge, no systematic review of the literature has been reported focusing on these cases. Thus, the aim of our study was to systematically review the literature on PSM in women with low- or intermediate-risk EC.

## 2. Materials and Methods

### 2.1. Study Protocol

A protocol was built a priori to systematically review the literature and was registered in the PROSPERO International Prospective Register of Systematic Reviews (ID: CRD42024508632). Each review stage was independently performed by two authors, and disagreements were solved by discussion with a third author. The whole study was reported following the Preferred Reporting Item for Systematic Reviews and Meta-analyses (PRISMA) statement and checklist [13].

### 2.2. Search Strategy

Using several combinations of topic-related words (i.e., “metastas*’’; ‘’recurren*’’; ‘’port’’; ‘’endometr*’’; “cancer”; “tumor’’; carcinoma’; ‘’neoplasia’’; ‘’malignancy’’), 6 electronic databases (i.e., Google Scholar, Web of Sciences, Scopus, MEDLINE, ClinicalTrial.gov, and EMBASE) were searched from their inception to January 2023 for all peer-reviewed studies which reported PSM in low- or intermediate-risk EC women. A priori defined exclusion criteria were: studies in languages other than English and literature reviews.

### 2.3. Risk of Bias within Studies Assessment

The risk of bias within studies was assessed by adopting: (1) the methodological quality and synthesis of case series and case reports [14] for case report and case series and (2) the Methodological Index for Non-Randomized Studies (MINORS) [15] for observational studies.

In detail, for case reports and case series, the following applicable domains were examined: (1) selection (i.e., did the patients represent the whole experience of the center?), (2) ascertainment (i.e., was the outcome adequately ascertained?), (3) causality (i.e., was PSM histologically confirmed? Was the follow-up length of at least 1 year?), and (4) reporting (i.e., were the cases described with sufficient details to allow other investigators to replicate the research or to allow practitioners to make inferences related to their own practice?).

For observational studies, the following seven applicable domains were analyzed: (1) a clearly stated aim (i.e., was the addressed aim relevant and precise?); (2) inclusion of consecutive patients (i.e., were all eligible patients included in the study during the study period?); (3) prospective collection of data (i.e., were data collected according to a protocol established before the beginning of the study?); (4) endpoints appropriate to the aim of the study (i.e., was endpoint appropriate to the aim considered?); (5) unbiased assessment of the study’s endpoints (i.e., was the assessment of study endpoints unbiased?); (6) follow-up period appropriate for the aim of the study (i.e., was the follow-up length of at least 1 year?); and (7) loss to follow-up less than 5% (i.e., were patients lost to follow-up less than 5%?).

The included studies were considered by the authors as having a “low risk”, “high risk”, or “unclear risk” of bias in each domain based on whether the data were “reported and adequate”, “reported but inadequate”, or “not reported”, respectively.

### 2.4. Data Extraction

Original data were extracted from the included studies without modification. In particular, we extracted data regarding the characteristics of the included studies (i.e., setting, study design, study period, sample size), patients (i.e., age, body mass index), treatment (surgical approach, surgical treatment and staging, port number, uterus and adnexa route of removal, lymph nodes route of removal, uterus manipulation, prevention strategies for port site metastasis, adjuvant therapy), primary tumor (i.e., histotype, International Federation of Gynaecology and Obstetrics (FIGO) grade, 2009 FIGO stage), PSM (PSM time from surgery, number of PSMs, PSM location, PSM size, concomitant metastasis, metastasis prior PSM, post-PSM relapse, local treatment for PSM, systemic treatment for PSM, disease-free survival (DFS) from PSM, overall survival (OS) from PSM detection, last negative follow-up known) [16].

We also collected and included in the data extraction a case of PSM from the medical records and electronic databases of the Gynecologic Oncology and Surgery Unit, Responsible Research Hospital, Campobasso, Italy. Data were reported in the tables to be comparable with those of cases from the studies included in the systematic review.

## 3. Results

### 3.1. Study Selection and Characteristics of the Included Studies

After the electronic database searches, 547 articles were identified. Duplicate removal resulted in 95 articles. Of these, 26 articles remained after title screening, and 24 remained after abstract screening. Of the 24 articles assessed for eligibility, 17 articles were excluded because they evaluated ovarian cancer, vulvar cancer, cervical cancer, high–intermediate-risk or high-risk EC. Finally, seven articles with 13 patients (including our case) were included in the systematic review (Figure 1).

Of the included studies, two were observational, retrospective, cohort studies; five were case reports (including our case); and one was a case series (Table 1).

### 3.2. Patients, Treatment, and Primary Tumor Characteristics

From the studies reporting information, patients’ ages and body mass indexes ranged from 45 to 73 years and from 38.5 to 40 kg/m^2^, respectively.

The EC had an endometrioid histotype in 76.9% (n = 10) of cases and serous in 23.1% of cases (n = 3). FIGO grade 1–2 was present in 76.9% (n = 10) of cases and 3 in 23.1% of cases (n = 3) cases, while 2009 FIGO stage IA was present in 84.6% of cases (n = 11) and IB in 15.4% of cases (n = 2).

The surgical approach was laparoscopic for 61.5% of cases (n = 8) and robotic for 38.5% (n = 5). Surgical treatment and staging consisted of total laparoscopic hysterectomy (TLH) + bilateral salpingo-ophorectomy (BSO) in 100% (n = 13) of cases, while lymph node dissection (LND) was performed in 61.5% (n = 8), sentinel lymph node (SLN) biopsy in 7.7% (n = 1), and washing in 30.8% (n = 4). The port number ranged from four to five. The route of removal for the uterus and adnexa was vaginal in 38.5% of patients (n = 5), transpubic mini-laparotomy in 7.7% (n = 1), and not reported in 53.8% (n = 7). The route of removal for the lymph nodes was assistant port with endobag in 15.4% of cases (n = 2), sovrapubic port (endobag use not reported) in 7.7% (n = 1), transpubic mini-laparotomy in 7.7% (n = 1), and not reported in 69.2% (n = 9). Uterus manipulation was performed in 30.8% (n = 4) of women, while it was not in 7.7% (n = 1) (data were missing for 61.5% (n = 8)). Prevention strategies for PSM were closure of the fascia in 38.5% (n = 5) of cases, irrigation with saline/povidone of the port site in 30.8% (n = 4), tubal coagulation in 15.4% (n = 2), and no uterine manipulation in 7.7% (n = 1).

Adjuvant treatment consisted of vaginal brachytherapy (7.7% (n = 1)), external beam radiation therapy (EBRT, one patient), chemotherapy (7.7% (n = 1)), oral micronized 17-b-estradiol (7.7% (n = 1)), EBRT + vaginal brachytherapy (15.4% (n = 2)), vaginal brachytherapy + chemotherapy (7.7% (n = 1)), and observation alone (38.5% (n = 5)) (Table 2).

PSM occurred 3–44 months after surgery and involved from 1 to 3 port sites (right lateral port in 53.8% (n = 7) of cases, left lateral port in 23.1% (n = 3), umbilical port in 7.7% (n = 1), and not reported in 15.4% (n = 2)). The PSM size ranged from 0.5–16 cm.

EC metastasis was concomitant in 30.8% (n = 4) of women (vaginal cuff in 7.7% (n = 1), right rectus abdominis muscle + bulky right common iliac lymph node in 7.7% (n = 1), pelvis in 15.4% (n = 2)), prior to PSM in 7.7% (n = 1, posterior virginal wall), and post-PSM in 30.8% (n = 4) (lung and brain in 7.7% (n = 1), abdomen and pelvic lymph node in 7.7% (n = 1), left pelvic side and psoas muscle in 7.7% (n = 1), right lateral port in 7.7% (n = 1), data missing in 15.4% (n = 2)).

PSM local treatment was reported in 76.9% (n = 10) of patients. It consisted of resection alone in 15.4% (n = 2) of cases, resection + RT in 46.1% (n = 6), and resection + laparotomy + RT in 15.4% (n = 2).

PSM systemic treatment was chemotherapy in 53.8% (n = 7) of women, while it was not performed in 30.8% (n = 4) and not reported in 15.4% (n = 2).

OS ranged from 0.6 to 2.2 years, while DSS ranged from 0.6 to 1.8 years (Table 3).

Our case of PSM was a 65-year-old woman with a body mass index of 40 kg/m^2^ who presented at the Gynecologic Oncology and Surgery Unit, Responsible Research Hospital, Campobasso, Italy, in October 2022 for postmenopausal abnormal uterine bleeding. After hysteroscopy with biopsies and imaging, she had a histological diagnosis of FIGO grade 1, stage IA, endometrial endometrioid carcinoma. The patient underwent laparoscopic surgery and staging for EC. A type A radical hysterectomy [25] with bilateral adnexectomy and sentinel lymph node biopsy [26] were performed. The histological examination reported a FIGO grade 1, endometrioid endometrial adenocarcinoma with “endocervical-type” aspect and infiltrating myometrium less than 50%, without lymphovascular space or node involvement. At immunohistochemical analysis, the neoplastic cells were positive for estrogen receptors, tumor protein 53 (p53) was completely absent, and there was a proficient expression of mismatch repair proteins. The 2009 FIGO stage was confirmed as IA. Three months after surgery, the patient had dehiscence of the umbilical wound: she complained of dark secretions, and a palpable black nodular lesion was detected at this level. Thorax–abdomen–pelvis computed tomography revealed the presence of pelvic lesions and a solid lesion (5 × 3 × 3 cm) at the umbilical scar level, in continuity with the underlying omentum and with another solid lesion (3 × 2 cm) which extended into the subcutaneous tissue, reaching the right rectus abdominis muscle. This finding was suggestive of a PSM. The biopsy with histological examination of the umbilical lesion confirmed the diagnosis of endometrial adenocarcinoma metastasis. Furthermore, a PET scan was performed, confirming the presence of the above-described findings. The patient was then referred for resection of the lesion and ongoing neoadjuvant chemotherapy with carboplatin and taxol.

### 3.3. Risk of Bias Assessment

The risk of bias within studies was assessed by adopting the methodological quality and synthesis of case series and case reports [14] for five studies [20,21,22,23,24], and the MINORS [15] for two studies [18,19].

In detail, of five studies [20,21,22,23,24] assessed through the methodological quality and synthesis of case series and case reports [14], one study was considered as having a low risk of bias in the “Selection” domain [24], while four studies were judged as having an unclear risk of bias because they did not report whether the case represented the whole experience of the center [20,21,22,23]. 

In the “Ascertainment” domain, one study was considered as having an unclear risk of bias because the outcome was not clearly stated [20], while the other studies were considered as having a low risk of bias [21,22,23,24].

In the “Causality” domain, two studies were considered as having an unclear risk of bias because the follow-up length was unclear [20,24], two studies were considered as having a high risk of bias because the follow-up length was not at least 1 year for all cases [21,23], and the remaining study was considered as having a low risk of bias [22].

In the “Reporting” domain, one study was considered as having an unclear risk of bias because the case was only partially described with sufficient details to allow other investigators to replicate the research [20]. The remaining studies were judged as having a low risk of bias [21,22,23,24] (Figure 2).

Regarding the two studies [18,19] assessed through MINORS [15], both were categorized as having a low risk of bias for the “A clearly stated aim”, “Inclusion of consecutive patients”, “Prospective collection of data”, Endpoints appropriate to the aim of the study”, and “Unbiased assessment of the study endpoint” domains [18,19]. For the “Follow-up” and “Loss to follow-up less than 5%” domains, one study was considered as having an unclear risk of bias because the follow-up length and the number of patients lost to follow-up were unclear [18], while the other study was judged as having a low risk of bias [19] (Figure 3).

## 4. Discussion

This study showed that, although rarely, PSM can occur in patients with a low- or intermediate-risk EC, even involving more than one port site and reaching large sizes (up to 16 cm). In particular, it can also occur in women with endometrioid histotype, FIGO grade 1–2, and 2009 FIGO stage IA EC, as well as independently from the endoscopic approach (laparoscopic or robotic), the lymph node staging type (LND or SLN biopsy), the number of ports, the route of specimen removal (assistant or sovrapubic port with endobag, or transpubic mini-laparotomy), and the prevention strategies (closure of the fascia, irrigation with saline/povidone of the port site, tubal coagulation, and no uterine manipulation). PSM can be isolated or associated with concomitant, prior, or subsequent metastases to other sites (e.g., vaginal cuff, right rectus abdominis muscle, iliac or pelvic lymph nodes, pelvis, psoas muscle, lung, and brain). PSM is locally treated in most cases, with resection alone, resection + RT, or resection + laparotomy + RT. Adjuvant treatment includes several options, such as vaginal brachytherapy, EBRT, chemotherapy, oral micronized 17-b-estradiol, EBRT + vaginal brachytherapy, vaginal brachytherapy + chemotherapy, or observation alone. In the reported cases, OS ranged from 0.6–2.2 years, while DSS ranged from 0.6–1.8 years.

In EC women, PSM has been described as rare and mainly concerning high-risk cases, with an overall incidence of 0.18–0.33% [4,11,12]. In low- or intermediate-risk EC women, PSM’s incidence is even lower. However, the incidence could be underestimated, probably due to the fact that only two cohort studies have been reported in the literature, and PSM can be considered less relevant in cases with concomitant metastases. To the best of our knowledge, overall, only 13 PSM cases (including the present case) have been reported in low- or intermediate-risk EC women to date.

The pathophysiology of PSM seems to be multifactorial and is not well understood yet. Among several factors, it has been hypothesized that CO_2_ insufflation may affect the spreading of tumor cells at the port site through the so-called “chimney effect” [27,28,29]. Nevertheless, data available in the literature appear controversial, with some studies reporting the same risk of tumor recurrence at the incision scar after laparoscopic and open surgery [30] and other studies reporting a three-times-increased incidence with the laparoscopic approach [18,31,32,33,34]. Moreover, although it has been reported that high intraperitoneal pressure leads to a higher chance of implantation of neoplastic cells according to the “chimney effect”, no difference in the risk of PSM was found between the robotic and the laparoscopic approach. In fact, robotic surgery allows for a lower pressure compared to conventional laparoscopy [4]. In addition, the robotic system might reduce the risk of contamination by tumor cells due to the possibility of inserting the instruments only once during surgery. In our systematic review, we found that PSM may occur independently from the endoscopic approach (laparoscopic or robotic). Regarding the localization of PSM, Palomba et al. found that neoplastic contamination of the port site appeared not to be correlated with direct contact of cancer cells [5]. In particular, of the 12 cases reported in the study by Palomba et al., the majority of PSMs were not identified at the extraction port: specifically, five PSMs were found in the left lateral port, three in the right lateral port, one in the sub-umbilical port, and two in the umbilical port (not extraction point), while localization was not reported in the remaining case [5]. In our systematic review, the extraction port was reported in only four cases (assistant port in two cases, sovrapubic port in one case, and transpubic mini-laparotomy in the remaining case); of these, the PSM’s location was not reported in two cases, while it was not at the extraction point in the other two cases. Despite the low number of cases, our data appear to be in accordance with Palomba et al.’s study; indeed, most of the PSMs were not found in the extraction port.

Concerning other factors associated with PSM, some tumor characteristics, such as histotype, grade, stage, and molecular signature, were assessed as possible risk factors. For instance, Lönnerfors et al. found a four-times-higher incidence of PSM in the retrieval port in more aggressive histotypes such as clear cell and carcinosarcoma [35], while Martinez et al. reported a higher incidence in advanced disease [4]. In a systematic review assessing PSM in EC patients of all risk groups, Palomba et al. reported heterogeneous data regarding the association between PSM incidence, EC stage, and histotype, probably due to the confounding effect of adjuvant treatments such as chemotherapy and EBRT, which could have included the laparoscopic ports in the radiation field [5]. Overall, nonisolated PSMs arose from EC with advanced stages, aggressive histotype, and FIGO grade 3 in 90%, 50%, and 60% of cases, respectively. Conversely, isolated PSMs arose from EC with early stages, FIGO grade 1–2, and stage IA in 75%, 76.9%, and 84.6% of cases, respectively [5].

Hewett et al. suggested that the direct implantation of tumor cells from contact with specimens or contaminated laparoscopic instruments might be another pathophysiologic mechanism of PSM. On this basis, Hewett et al. recommended the removal of all ports when the peritoneal cavity is insufflated and to wipe all the instruments with cytotoxic agents [36]. Moreover, several other prevention strategies for PSM, such as closure of the fascia, irrigation with saline/povidone of the port site, tubal coagulation, and no uterine manipulation, have been proposed over time [4,24,33,37,38,39,40,41]. In our systematic review, we found PSM despite the adoption of these prevention strategies. This might indicate that the direct implantation of tumor cells from the specimens or contaminated laparoscopic instruments is not a real pathophysiologic mechanism for PSM or that these prevention strategies are not effective, highlighting the need for novel prevention strategies. Similarly, the lymph node staging approach (LND or SLN biopsy), the number of ports, and the routes of specimen removal (assistant or sovrapubic port with endobag, or transpubic mini-laparotomy) seemed not to affect the risk of PSM. Another prevention strategy may be strict patient selection. Indeed, it was demonstrated that, in ovarian cancer patients with advanced disease, high-grade histology, positive lymph nodes, and ascites have a higher risk of PSM [42]. Therefore, patients with higher tumor aggressiveness may not be ideal candidates for minimally invasive procedures. However, additional studies are necessary to confirm these findings.

It has been described that surgical complications might also contribute to the development of PSM. Among these, in the cases included in our study, only one intraoperative surgical complication possibly associated with PSM was described. In detail, in one case, the uterus was perforated during the manipulation, with possible tumor dissemination [23].

Regarding the presence of concomitant metastasis, Palomba et al. found the notion that PSM could be considered as an isolated disease and not as an expression of an underlying micrometastatic disease to be questionable [5]. In fact, in all five cases of non-isolated PSM with available follow-up, the patients died after an average of 10 months after the recurrence. Similarly, in the isolated PSM sample, only one out of three patients was alive and disease-free at 10 months of follow-up. Our systematic review found four concomitant metastases, one metastasis prior the onset of the PSM, and four metastases after the treatment of the PSM. Of the four cases with non-isolated PSM, only one was disease-free at 9 months of follow-up. 

For the treatment of PSM, there is no consensus in the literature. In our systematic review, PSM local treatment was reported in 76.9% (n = 10) of patients and consisted of resection alone in 15.4% (n = 2) of cases, resection + RT in 46.1% (n = 6), and resection + laparotomy + RT in 15.4% (n = 2). The systemic treatment for PSM was chemotherapy in 53.8% (n = 7) of women, while it was not performed in 30.8% (n = 4) and was not reported in 15.4% (n = 2). According to Palomba et al., excision followed by RT can be an appropriate treatment for isolated PSM, while the role of chemotherapy might be more supported in non-isolated PSM [5]. In a retrospective observational cohort study, Grant et al. reported favorable outcomes for PSM treated with resection and RT with or without chemotherapy. In detail, almost half of the patients remained disease-free after 2 years [19]. In contrast to these findings, Palomba et al. found a poor prognosis in patients with both isolated and non-isolated PSM, although it was unclear whether a PSM defined as isolated could hide a disseminated micrometastatic pathology [5]. In the present systematic review, only Grant et al. reported data on DFS [19], OS, and median interval time between the laparoscopic treatment for the primary tumor and the radiographic diagnosis of PSM. Unfortunately, drawing comprehensive conclusions appears not to be feasible with only one source providing these outcomes. Further studies are necessary for more reliable survival estimates.

Although, to the best of our knowledge, our study may be the first systematic review focusing on PSM in low- or intermediate-risk EC women, the generalizability of our findings might be limited by the low number of cases (n = 13), the study design (case reports; case series; and observational, retrospective, cohort studies) of the included studies, and the variability of the reported outcomes and variables. Due to the rarity of the condition, it appears difficult to build interventional prospective studies.

## 5. Conclusions

Although rarely, PSM can occur in patients with low- or intermediate-risk EC, independently of tumor histologic characteristics, endoscopic approach, lymph node staging type, number and site of the port, route of specimens removal, prevention strategies for PSM, and concomitant metastases. Despite the lack of consensus in literature, resection and RT with or without chemotherapy seems to be the most appropriate treatment for EC women with PSM. Several prevention strategies have been described, such as closure of the fascia, irrigation with saline/povidone of the port site, tubal coagulation, no uterine manipulation, and strict selection of patients; however, their efficacy is unclear, and novel prevention strategies are needed. OS and DFS appear poor in these patients. Additional cases are needed to confirm and further explore this rare EC metastasis.

## Figures and Tables

**Figure 1 cancers-16-02682-f001:**
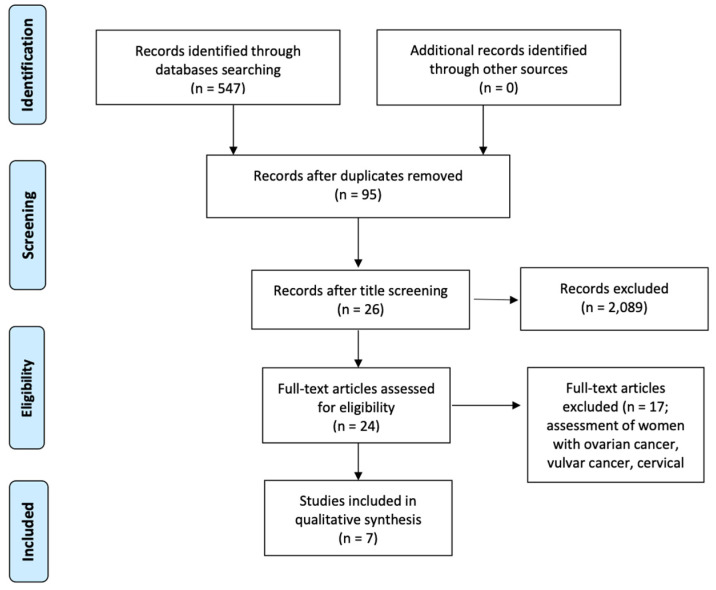
Flowchart of study selection for the systematic review and meta-analysis (PRISMA (Preferred Reporting Item for Systematic Reviews and Meta-analyses) template). From [17]. For more information, visit https://www.prisma-statement.org/ (accessed on 1 July 2024).

**Figure 2 cancers-16-02682-f002:**
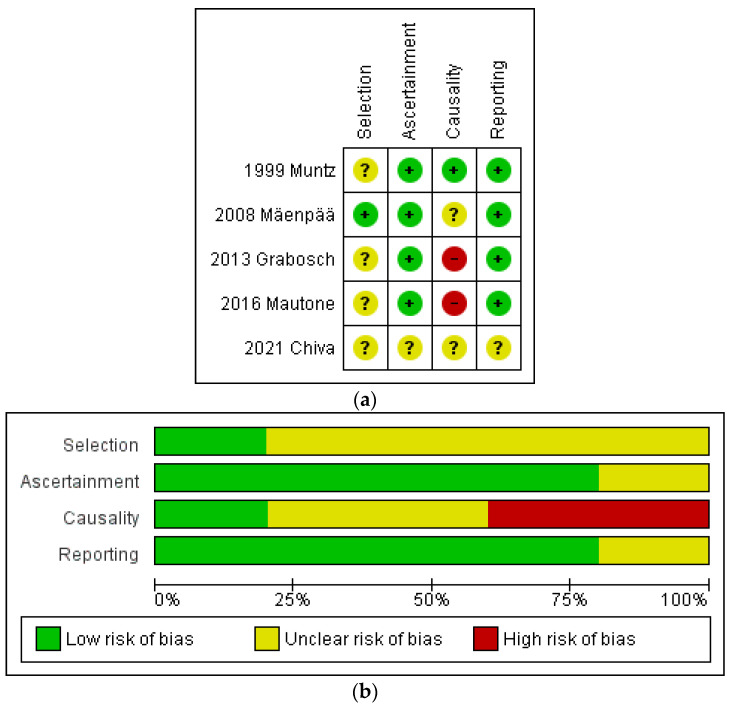
(**a**) Assessment of risk of bias through the methodological quality and synthesis of case series and case reports [20,21,22,23,24]. Summary of risk of bias for each study: Plus sign: low risk of bias; minus sign: high risk of bias; question mark: unclear risk of bias. (**b**) Risk of bias graph, with each risk of bias item presented as a percentage across all included studies.

**Figure 3 cancers-16-02682-f003:**
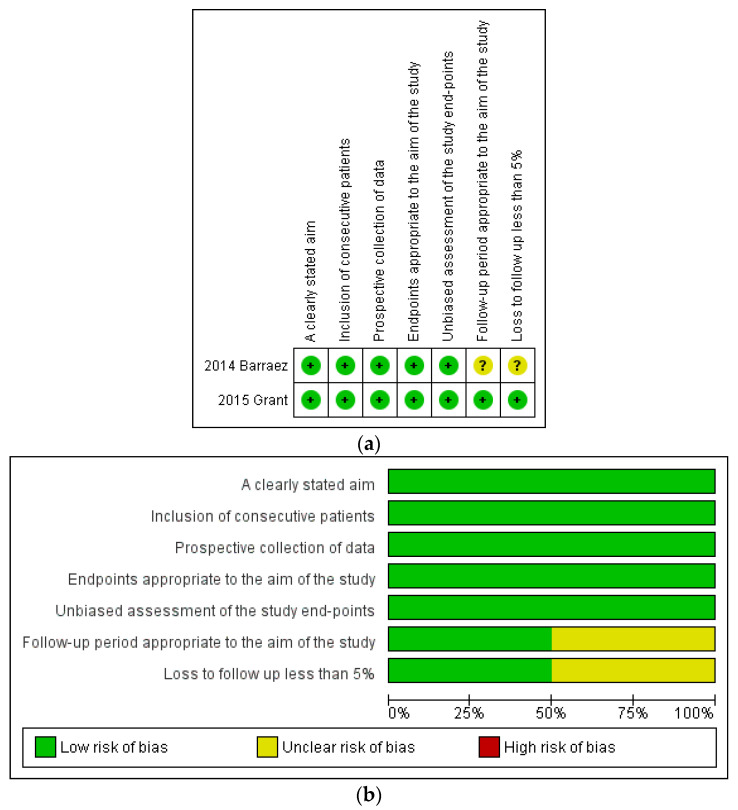
(**a**) Assessment of risk of bias through the Methodological Index for Non-Randomized Studies (MINORS) [18,19]. Summary of risk of bias for each study: Plus sign: low risk of bias; question mark: unclear risk of bias. (**b**) Risk of bias graph with each risk of bias item presented as a percentage across all included studies.

**Table 1 cancers-16-02682-t001:** Characteristics of the included studies.

Study	Setting	Study Design	Study Period (Month)	Sample Size
2015 Barraez [18]	-	Observational retrospective cohort study	60	2
2015 Grant [19]	Anderson Cancer Center, Texas	Observational retrospective cohort study	204	4
2021 Chiva [20]	Clinica Universitad de Navarra Madrid, Spain	Case report	NA	1
2016 Mautone [21]	University Hospital of Parma, Italy	Case report	NA	1
1999 Muntz [22]	Virginia Mason Medical Center, USA	Case report	NA	1
2013 Grabosch [23]	St. Mary’s Health Center, St. Louis, Missouri	Case series	NA	2
2009 Mäenpää [24]	University Hospital of Tampere, Tampere, Finland	Case report	NA	1
Present case	Responsible Research Hospital, Campobasso, Italy	Case report	NA	1
TOTAL	-	-	-	13

-: not available; NA: not applicable.

**Table 2 cancers-16-02682-t002:** Patients, treatment, and primary tumor characteristics.

Study	Age [Years]	BMI [Kg/m^2^]	Approach	Histotype	FIGO Grade	FIGO Stage	Surgical Treatment and Staging	Port n	Uterus and Adnexa Route of Removal	Lymph Nodes Route of Removal	Uterus Manipulation	Prevention Strategies for Port Site Metastasis	Adjuvant Therapy
2015 Barraez [18]	65	38.5	R	E	3	IA	TLH + BSO + lymph node staging (LND/not specified)	5	Vaginal (without endobag)	Assistant port with endobag	-	Closure of the fascia +irrigation of the port site with saline	VBT
	67	40	R	E	2	IB	TLH + lymph node staging (LND/not specified)	5	Vaginal (without endobag)	Assistant port with endobag	-	Closure of the fascia +irrigation of the port site with saline	EBRT
2015 Grant [19]	63	-	L	E	2	IA	TLH + BSO	-	-	-	-	-	Observation
	45	-	L	E	2	IA	TLH + BSO	-	-	-	-	-	EBRT + VBT
	60	-	L	S	3	IA	TLH + BSO + LND	-	-	-	-	-	VBT + CT
	73	-	R	S	3	IA	TLH + BSO + LND	-	-	-	-	-	CT
2021 Chiva [20]	73	-	L	S	3	IA	TLH + BSO + pelvic and paraaortic LND	-	-	-	-	-	EBRT + VBT
2016 Mautone [21]	57	-	L	E	1	IA	TLH + BSO + washing	4	Vaginal (without endobag)	-	Yes	Tubal coagulation Irrigation with povidone	Ob
1999 Muntz [22]	58	-	L	E	2	IA	TLH + BSO + pelvic LND + washing	4	Vaginal (endobag use not reported)	Sovrapubic port (endobag use not reported)	Yes	Fascia suture of 10 mm ports	OME
2013 Grabosch [23]	56	-	R	E	1	IA	TLH + BSO + LND	-	-	-	-	-	Ob
	54		R	E	1–2	IA	TLH + BSO + pelvic and para-aortic LND + washing	-	-	-	Yes (perforation)	-	Ob
2009 Mäenpää [24]	68		L	E	2	IB (IC)	TLH + BSO + washing	4	Vaginal (difficult due to uterine leiomyoma)	-	Yes	Tubal coagulation Fascia suture of 10–12 mm ports	EBRT
Present case	65	40	L	E	1	IA	TLH + BSO + SLN biopsy	4	Transpubic mini-laparotomy	Transpubic mini-laparotomy	No	No uterine manipulation + 5% povidone-iodine; closed using a monofilament suture	Ob
TOTAL	45–73	38.5–40	L (n = 8)	E (n = 10)	1–2 (n = 10)	IA (n = 11)	TLH + BSO (n = 13)	\	Vaginal (n = 5)	Assistant port with endobag (n = 2)	Yes (n = 4)	Closure of the fascia (n = 5)	VBT (n = 1)
			R (n = 5)	S (n = 3)	3 (n = 3)	IB (n = 2)	LND (n = 8)		Transpubic mini-laparotomy (n = 1)	Sovrapubic port (endobag use not reported) (n = 1)	No (n = 1)	Irrigation of the port site with saline/povidone (n = 4)	EBRT (n = 1)
							SLN biopsy (n = 1)		Not reported (n = 7)	Transpubic mini-laparotomy (n = 1)	Not reported (n = 8)	Tubal coagulation (n = 2)	CT (n = 1)
							Washing (n = 4)			Not reported (n = 9)		No uterine manipulation (n = 1)	OME (n = 1)
													EBRT + VBT (n = 2)
													VBT + CT (n = 1)
													Ob (n = 5)

-: not reported; R: robotic approach; L: laparoscopic approach; E: endometrioid histotype; S: serous histotype; TLH: total laparoscopic hysterectomy; BSO: bilateral salpingo-ophorectomy; LND: lymph node dissection; EBRT: external beam radiation therapy; SLN: sentinel lymph node; CT: chemotherapy; VBT: vaginal brachytherapy; OME: oral micronized estradiol; Ob: observation alone.

**Table 3 cancers-16-02682-t003:** Port site metastasis characteristics.

Study	PSM Time from Surgery (Months)	PSM n	PSM Location	PSM Size (cm)	Concomitant Metastasis	Metastasis Prior PSM	Post PSM Relapse	Local Treatment for PSM	Systemic Treatment for PSM	DFS from PSM (Years)	OS from Detection (Years)	Last Negative Follow-Up Known (Month)
2015 Barraez [18]	9	-	-	-	Vaginal cuff	None	-	-	-	-	-	-
	19	-	-	-	None	None	-	-	-	-	-	-
2015 Grant [19]	10	1	Right lower abdomen	2.9	None	None	None	Resection + 3DCT	None	0.6	0.6	-
	44	1	Left rectus abdominis	9	None	Posterior virginal wall	Lung and brain	Resection + IMRT	None	1.42	2.6	-
	12	1	Right anterior abdomen	0.9	None	None	None	Resection + electrons (9 meV)	None	1.5	1.5	-
	15	3	Right abdomen	1, 0.5, 0.5	None	None	Abdomen, pelvic lymph node	Resection + 3DCT (4 field)	Letrozole	1.83	2.2	-
2021 Chiva [20]	6	1	Right iliac fossa	16 × 12	None	None	None	Resection	NACT+ ACT + moAb	-	-	-
2016 Mautone [21]	7	1	Right iliac fossa near the right rectum	2.8 × 2.4 × 1.9	Right rectus abdominis muscle + Bulky right common iliac lymph node	None	None	Resection + RT	CT	-	-	9
1999 Muntz [22]	21	1	Left rectus muscle	5	None	None	Left pelvic side and psoas muscle *	Resection + Removal of residual pelvic lymph nodes, PALND, biopsy of the vaginal cuff, infracolic omentectomy, peritoneal biopsies, peritoneal and diaphragmatic cytology. + RT	None	-	-	30
2013 Grabosch [23]	10	1	Right lower quadrant	3	None	None	None	Resection + RT	CT	-	-	13
	36	1	Left lower quadrant		None	None	None	Resection + RT	CT	-	-	4
2009 Mäenpää [24]	6	1	Right lateral port	5.7 × 2.2 × 3.2	Pelvis	None	Right lateral port	Resection + laparotomy + RT	NACT + MPA	-	-	-
Present case	3	1	Umbilical port and underling omentum	5 × 3 × 3	Pelvis	None	None	Resection	NACT	-	-	-
TOTAL	3–44	1–3	Right lateral port (n = 7)	0.5–16	Vaginal cuff (n = 1)	Posterior virginal wall (n = 1)	Lung and brain (n = 1)	Resection + RT (n = 7)	CT (n = 7)	0.6–1.83	0.6–2.2	4–13
			Left lateral port (n = 3)		Right rectus abdominis muscle + Bulky right common iliac lymph node (n = 1)	None (n = 12)	Abdomen, pelvic lymph node (n = 1)	Resection alone (n = 2)	None (n = 4)			
			Umbilical port (n = 1)		Pelvis (n = 2)		Left pelvic side and psoas muscle * (n = 1)	Resection + laparotomy + RT (n = 2)	- (n = 2)			
			Not reported (n = 2)		None (n = 9)		Right lateral port (n = 1)	- (n = 2)				
							None (n = 7)					
							Not reported (n = 2)					

-: not reported; RT: radiotherapy; PSM: port site metastasis; 3DTC: dimensional conformal therapy; IMRT: intensity-modulated radiation therapy; DFS: disease-free survival; OS: overall survival; PALND: paraaortic lymph node dissection; CT: chemotherapy; NACT: neoadjuvant chemotherapy; ACT: adjuvant chemotherapy; moAb: monoclonal antibodies, MPA: medroxyprogesterone acetate, *****: information reported by Palomba et al. 2012 [5].

## Data Availability

The data that support the findings of this study are available upon request from the corresponding author.

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
