# Peer review of "Port Site Metastasis in Women with Low- or Intermediate-Risk Endometrial Carcinoma: A Systematic Review of Literature"

_cancers, 2024, doi:10.3390/cancers16152682_

Round 1

Reviewer 1 Report

Comments and Suggestions for Authors

This is an interesting paper reviewing the literature regarding the treatment and incidence of port site metastases in relatively low risk endometrial cancers.  

As expected the frequency is very low.  There are only 13 cases.  It is hard to make a recommendation on this but the authors have done their best.

The conclusions seem reasonable.  Excision and possible chemo or radiation.

Comments on the Quality of English Language

This is an interesting paper reviewing the literature regarding the treatment and incidence of port site metastases in relatively low risk endometrial cancers.  

As expected the frequency is very low.  There are only 13 cases.  It is hard to make a recommendation on this but the authors have done their best.

The conclusions seem reasonable.  Excision and possible chemo or radiation.

I think that the impact is somewhat low but 

Author Response

We thank the Reviewer for the kind comments

Reviewer 2 Report

Comments and Suggestions for Authors

In table 2, one case that is serous endometrial cancer is grade 2, it may be grade 3 (Chiva 2021).

Aim of this study is to evaluate PSM in patients with low and intermediate risk endometrial cancer. However there are 3 high-risk patients (serous endometrial cancer)

It is challenging to evaluate risk factors of PSM at 13 cases. Low case number  is main weakness of this study.

Author Response

Comment #1: In table 2, one case that is serous endometrial cancer is grade 2, it may be grade 3 (Chiva 2021).

Response: We thank the Reviewer for the comment. We corrected the typo in Table 2.

Comment #2: Aim of this study is to evaluate PSM in patients with low and intermediate risk endometrial cancer. However there are 3 high-risk patients (serous endometrial cancer)

Response: We thank the Reviewer for the comment. In fact, the cases of serous (non-endometrioid) endometrial cancer are stage IA. Such cases have an intermediate risk for the ESGO-ESTRO-ESP guidelines for endometrial cancer [Concin N, Matias-Guiu X, Vergote I, Cibula D, Mirza MR, Marnitz S, Ledermann J, Bosse T, Chargari C, Fagotti A, Fotopoulou C, Gonzalez Martin A, Lax S, Lorusso D, Marth C, Morice P, Nout RA, O'Donnell D, Querleu D, Raspollini MR, Sehouli J, Sturdza A, Taylor A, Westermann A, Wimberger P, Colombo N, Planchamp F, Creutzberg CL. ESGO/ESTRO/ESP guidelines for the management of patients with endometrial carcinoma. Int J Gynecol Cancer. 2021 Jan;31(1):12-39. doi: 10.1136/ijgc-2020-002230. Epub 2020 Dec 18. PMID: 33397713]

Comment #3: It is challenging to evaluate risk factors of PSM at 13 cases. Low case number  is main weakness of this study.

Response: We thank the Reviewer for the comment. We added this limitation in the discussion section.

Reviewer 3 Report

Comments and Suggestions for Authors

The authors present a manuscript which aims to clarify the risk factors and optimal treatment for port site metastasis occurring in women with low or intermediate risk endometrial cancer. The study has been conducted properly and the manuscript is well written. However, several corrections should be made to achieve better comprehension. First, the whole manuscript should be edited by a native English speaker who is familiar with medical writing so that typographical and grammatical errors should be corrected. Second, the authors should rewrite the conclusion part and more openly discuss about the clinical implications of their findings (i.e. how to prevent port site metastasis). Third, the references that were published before 2008 (except those included in the meta-analysis) should be replaced with newer and more up-to-date ones if possible. I would recommend that this manuscript can be accepted for publication in Cancers after required corrections have been made.

Comments on the Quality of English Language

The authors present a manuscript which aims to clarify the risk factors and optimal treatment for port site metastasis occurring in women with low or intermediate risk endometrial cancer. The study has been conducted properly and the manuscript is well written. However, several corrections should be made to achieve better comprehension. First, the whole manuscript should be edited by a native English speaker who is familiar with medical writing so that typographical and grammatical errors should be corrected. Second, the authors should rewrite the conclusion part and more openly discuss about the clinical implications of their findings (i.e. how to prevent port site metastasis). Third, the references that were published before 2008 (except those included in the meta-analysis) should be replaced with newer and more up-to-date ones if possible. I would recommend that this manuscript can be accepted for publication in Cancers after required corrections have been made.

Author Response

Comment #0: The authors present a manuscript which aims to clarify the risk factors and optimal treatment for port site metastasis occurring in women with low or intermediate risk endometrial cancer. The study has been conducted properly and the manuscript is well written. However, several corrections should be made to achieve better comprehension.

Response: We thank the Reviewer for the kind comment.

Comment #1: First, the whole manuscript should be edited by a native English speaker who is familiar with medical writing so that typographical and grammatical errors should be corrected.

Response: We thank the Reviewer for the comment. We revised the whole manuscript with a native English speaker to correct typographical and grammatical errors.

Comment #2: Second, the authors should rewrite the conclusion part and more openly discuss about the clinical implications of their findings (i.e. how to prevent port site metastasis). 

Response: We thank the Reviewer for the comment. We expanded the conclusion including strategies to prevent port sites metastasis. We also added and discussed another possible strategy to prevent PSM in the discussion section.

Comment #3: Third, the references that were published before 2008 (except those included in the meta-analysis) should be replaced with newer and more up-to-date ones if possible.

Response: We thank the Reviewer for the comment. We replaced the references published before 2008 with newer ones. Unfortunately, due to the scarcity of literature about port site metastases in endometrial carcinoma, in some cases we were not able to find more recent data and had to keep references published before 2008.

Comment #4: I would recommend that this manuscript can be accepted for publication in Cancers after required corrections have been made.

Response: We thank the Reviewer for the kind comment. 

Round 2

Reviewer 2 Report

Comments and Suggestions for Authors

You are right about that stage Ia non-non endometrioid cancer may be low risk for some societies. However, all non-endometriod cancer is accepted ay high risk pathology in FIGO 2023 staging system.